# EFFICIENT DISCRETE MULTI-MARGINAL OPTIMAL TRANSPORT REGULARIZATION

**Ronak Mehta**[*]  **Jeffery Kline**  **Vishnu Suresh Lokhande**  **Glenn Fung**  **Vikas Singh**
UW-Madison      Affirm        UW-Madison            Liberty Mutual   UW-Madison

## ABSTRACT

Optimal transport has emerged as a powerful tool for a variety of problems in machine learning, and it is frequently used to enforce distributional constraints. In this context, existing methods often use either a Wasserstein metric, or else they apply concurrent barycenter approaches when more than two distributions are considered. In this paper, we leverage multi-marginal optimal transport (MMOT), where we take advantage of a procedure that computes a generalized earth mover's distance as a sub-routine. We show that not only is our algorithm computationally more efficient compared to other barycentric-based distance methods, but it has the additional advantage that gradients used for backpropagation can be efficiently computed during the forward pass computation itself, which leads to substantially faster model training. We provide technical details about this new regularization term and its properties, and we present experimental demonstrations of faster runtimes when compared to standard Wasserstein-style methods. Finally, on a range of experiments designed to assess effectiveness at enforcing fairness, we demonstrate our method compares well with alternatives.

## 1 INTRODUCTION

The use of Optimal transport (OT) is now prevalent in many problem settings including information retrieval (Balikas et al., 2018; Yurochkin et al., 2019), image processing (Bonneel et al., 2014), statistical machine learning, and more recently, for ethics and fairness research (Kwegyir-Aggrey et al., 2021; Lokhande et al., 2020a). OT is well-suited for tasks where dissimilarity between two or more probability distributions must be quantified; its success was made possible through dramatic improvements in algorithms (Cuturi, 2013; Solomon et al., 2015) that allow one to efficiently optimize commonly used functionals. In practice, OT is often used to estimate and minimize the distance between certain (data-derived) distributions, using an appropriately defined loss functional. When one seeks to operate on more than two distributions, however, newer constructions are necessary to effectively estimate distances and transports. To this end, a well studied idea in the literature is the "barycenter," identified by minimizing the pairwise distance between itself and all other distributions given. The $d$-dimensional proxy distance is then defined as the sum of the distances to the barycenter.

**Computing barycenters.** Assuming that a suitably regularized form of the optimal transport loss is utilized, the pairwise distance calculation, by itself, can be efficient – in fact, in some cases, Sinkhorn iterations can be used (Cuturi, 2013). On the other hand, to minimize distances to the mean, most algorithms typically operate by repeatedly estimating the barycenter and those pairwise distances, and using a "coupling" strategy to push points toward the barycenter, or in other cases, summing over all pairwise distances. As the number of distributions grows, robustness issues can exacerbate (Alvarez-Esteban et al., 2008) and the procedure is expensive (e.g., for 50 distributions, 50 bins).

**A potential alternative.** Multi-marginal optimal transport (MMOT) is a related problem to the aforementioned task but to some extent, the literature has developed in parallel. In particular, MMOT focuses on identifying a joint distribution such that the marginals are defined by the input distributions over which we wish to measure the dissimilarity. The definition naturally extends the two-dimensional formulation, and recent work has explored a number of applications (Pass, 2015). But the MMOT computation can be quite difficult, and only very recently have practical algorithms been identified

---

[*]Corresponding Author, `ronakrm@cs.wisc.edu`

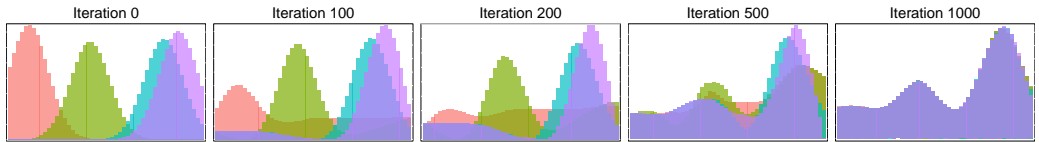

Figure 1: Starting and ending state of minimizing a multi-marginal OT distance. Each iteration minimizes the generalized Earth Mover's objective, and then updates each histogram in the direction provided by the gradient.

(Lin et al., 2022). Additionally, even if a suitable method for computing an analogous measure of distance were available, *minimizing* this distance to reduce dissimilarity (push distributions closer to each other) is practically hard if standard interior point solvers are needed just to compute the distance itself.

**Why and where is dissimilarity important?** Enforcing distributions to be similar is a generic goal whenever one wishes some outcome of interest to be agnostic about particular groups within the input data. In applications where training deep neural network models is needed, it is often a goal to enforce distribution similarity on model outputs. For example, in Jiang et al. (2020), the authors define fairness measures over the probability of the prediction, given ground truth labels. However, these methods are rarely extended to continuous measures among internal neural network activations, mainly due to the strong distributional assumptions needed (product of Gaussians) and the added algorithmic complexity of estimating the barycenter. These issues limit application of these ideas to only the final outputs of neural network models, where the distribution is typically binomial or multinomial. MMOT solutions might be employed here, but suffer similar computational limitations.

**Contributions. (1)** We identify a particular form of the discrete multi-marginal optimal transport problem which admits an extremely fast and numerically robust solution. Exploiting a recent extension of the classical Earth Movers Distance (EMD) to a higher-dimensional Earth Mover's objective, we show that such a construction is equivalent to the discrete MMOT problem with Monge costs. **(2)** We show that minimization of this *global* distributional measure leads to the harmonization of input distributions very similar in spirit to the minimization of distributions to barycenters (see Figure 1). **(3)** We prove theoretical properties of our scheme, and show that the gradient can be read directly off from a primal/dual algorithm, alleviating the need for computationally intense pairwise couplings needed for barycenter approaches. **(4)** The direct availability of the gradient enables a specific neural network instantiation, and with a particular scaffolding provided by differentiable histograms, we can operate directly on network activations (anywhere in the network) to compute/minimize the d-MMOT. We establish via experiments that computing gradients used in backpropagation is fast, due to rapid access to solutions of the dual linear program. We compare with barycenter-like approaches in several settings, including common fairness applications.

## 2 RELATED WORK

Despite originating with (Monge, 1781), optimal transport continues to be an active area of research (Villani, 2009). The literature is vast, but we list a few key developments.

**Early applications.** Starting in (Peleg et al., 1989), the idea of shifting "mass" around within an image was used for comparing images to each other and applied to image retrieval (Rubner et al., 2000), where the term "Earth Mover's Distance" (EMD) was introduced. EMD has since been widely used in computer vision: e.g., for image warping (Zhang et al., 2011), in supervised settings (Wang & Guibas, 2012), matching point sets (Cabello et al., 2008) and in scenarios involving histogram comparisons (Ling & Okada, 2007; Wang & Guibas, 2012; Haker et al., 2004).

**Modern machine learning.** The continuous optimal transport problem (Monge-Kantorovich problem), was originally presented in (Kantorovich, 1942; Kantorovitch, 1958). While the continuous problem has been studied intensively (Villani, 2021), uses of optimal transport within machine learning were possible due to (Cuturi, 2013), which showed that entropic regularization enables fast algorithms for EMD (two distributions with discrete support), and contributed to the success of Wasserstein distances in applications. Consequently, problems including autoencoders (Tolstikhin et al., 2018), GANs (Arjovsky et al., 2017), domain adaptation (Courty et al., 2016), word embeddings (Huang et al., 2016) and classification tasks (Frogner et al., 2015) have benefited via the use of optimal transport.

**Multi-marginal optimal transport.** Extending optimal transport theory to an *arbitrary number* of distributions has been studied on the theoretical side (Pass, 2015), and practical extensions have been proposed (Lin et al., 2022). But implementations that integrate directly into machine learning pipelines rely on either heuristics or modifications that result in an approximation of the original MMOT problem (Cao et al., 2019).

**Wasserstein barycenters.** One use case of our algorithms will be in formulations that involve Wasserstein barycenters. Given a set of probability distributions, the Wasserstein barycenter minimizes the *mean* of Wasserstein distances to *each* probability distribution in the set: a practical definition of the mean under the transportation distance (Luise et al., 2019; Cuturi & Doucet, 2014; Agueh & Carlier, 2011; Janati et al., 2020). Applications of Wasserstein barycenter include texture analysis (Rabin et al., 2011), sensor data fusion (Elvander et al., 2020), shape interpolation (Solomon et al., 2015), coupling problems (Rüschendorf & Uckelmann, 2002) and others (Ho et al., 2017). Very recently, polynomial time algorithms have been derived (Altschuler & Boix-Adsera, 2021).

**Fairness.** Proposals such as (Jiang et al., 2020; Chzhen et al., 2020; Gordaliza et al., 2019; Lokhande et al., 2022b) have all regularized models towards outcomes which have equal predictive power over subgroups within a population, measured using optimal transport distance. Informally, the idea is to operate on distributions that are supported on disparate groups, and ask that the distributions get "pushed" towards a common central distribution. This requires solving an optimal transport problem. Kwegyir-Aggrey et al. (2021)

**Greedy algorithms and extensions.** Hoffman et al. (1963) observed that there exists a family of linear-time greedy algorithms that solve the classical two-dimensional transportation problem. Later, Bein et al. (1995) extended the relevant definitions and the greedy algorithm to $d$-dimensional transportation problems. More recently, the results in Kline (2019), which we will use here, further extended this result to the dual program, and several theoretical properties of the generalized Earth Mover's problem were shown. A slightly different generalized $d$-dimensional Earth Mover's problem is explored in Erickson (2020), with a focus on statistical generalization properties.

## 3 BACKGROUND

Denote by $[n] := \{1, \ldots, n\}$, the set of positive integers no larger than $n$. For elements $x \in \mathbb{R}^n$, we denote the $i$th entry of $x$ as $x(i)$, e.g., for any $x \in \mathbb{R}^n$, $x = (x(i) : i \in [n])$. The positive orthant of $\mathbb{R}^n$ is denoted $\mathbb{R}^n_+ := \{x \in \mathbb{R}^n : x(i) \geq 0, i \in [n]\}$. We denote by $e := (1, \ldots, 1) \in \mathbb{R}^n$ the constant vector. For $q \geq 1$, we define the $q$-norm as $\|x\|_q := \left(\sum_{i \in [n]} |x(i)|^q\right)^{1/q}$, and if $q$ is suppressed, then $\|x\| := \|x\|_2$. A discrete probability distribution is a point $p \in \mathbb{R}^n_+$ with $e'p = \|p\|_1 = 1$.

Given a pair of discrete probability distributions $p_1, p_2 \in \mathbb{R}^n_+$, we may want to quantify similarity. Often, we do this by selecting from many measures, including the $q$-norm, KL-divergence or the Earth Mover's Distance (EMD). The EMD for a pair of distributions has several equivalent interpretations. First, let $p_1$ be a source of mass, and $p_2$ be a sink for mass, and $x(i,j)$, where $x \in \mathbb{R}^{n \times n}$, represent the flow of mass from $p_1(i)$ to $p_2(j)$. Denote by $c(i,j)$ the cost of moving one unit of mass from $p_1(i)$ to $p_2(j)$. The EMD between $p_1$ and $p_2$ is the minimal cost to transform $p_1$ into $p_2$, written as a linear program (LP):

$$\min_{x \in \mathbb{R}^{n \times n}_+} \sum_{i,j} c(i,j)x(i,j) \quad \text{s.t.} \quad \sum_j x(i,j) = p_1(i); \sum_i x(i,j) = p_2(j), \ (\forall i, j \in [n]). \quad (1)$$

The source-sink interpretation is asymmetric in $p_1$ and $p_2$, but the LP is symmetric in $p_1$ and $p_2$. It can be shown that the objective value of this LP defines a *metric* (Kantorovich, 1960), and the *optimal* value of the objective function can be interpreted as a distance between $p_1$ and $p_2$, and useful to quantify dissimilarity between pairs of distributions. In particular, $p_1 = p_2$ if and only if the optimal objective value of the Earth Mover's problem vanishes. The LP in (1) has an equivalent dual LP,

$$\max_{z_1, z_2 \in \mathbb{R}^n} z_1^\top p_1 + z_2^\top p_2 \quad \text{s.t.} \quad z_1(i) + z_2(j) \leq c(i,j), \ (\forall i, j \in [n]). \quad (2)$$

By strong duality, the optimal value of the primal program (1) equals the optimal value of the dual program (2). Many practical relaxations have been proposed for (1), including entropic regularization (Cuturi, 2013). Computation of the EMD is readily available, as in the Python Optimal Transport (POT) library (Flamary et al., 2021).

## 3.1 Discrete Multi-Marginal Optimal Transport

The foregoing approach applies only to $d = 2$ distributions, namely $p_1$ and $p_2$. We briefly review the extension to $d > 2$ distributions; the literature calls this *multi-marginal optimal transport (MMOT)*.

**Definition 3.1** (Discrete Multi-Marginal Optimal Transport (d-MMOT)). Let $p_1, \ldots, p_d \in \mathbb{R}_+^n$ be discrete probability distributions. Let $C_d : \mathbb{R}^{n^d} \to \mathbb{R}_+$. The discrete multi-marginal optimal transport problem (d-MMOT) can be written as

$$\min_{X \in \mathbb{R}^{n \times \cdots \times n}} C_d(X) \quad \text{s.t.} \quad X_i = p_i, \ (\forall i \in [d]),$$

where $X_i \in \mathbb{R}^n$ is the $i$-th marginal of $X \in \mathbb{R}^{n \times \cdots \times n} = \mathbb{R}^{n^d}$.

Following the original formulation (Kantorovich, 1942), we will restrict the cost function $C_d(\cdot)$ to the linear map, $C_d(X) := \langle c, X \rangle_\otimes$, where $c \in \mathbb{R}_+^{n \times \cdots \times n}$ is nonnegative. Here, the d-MMOT is the LP,

$$\min_{x \in \mathbb{R}_+^{n^d}} \sum_{i_1, \ldots, i_d} c(i_1, \ldots, i_d) \, x(i_1, \ldots, i_d) \quad \text{s.t.} \sum_{i_2, \ldots, i_d} x(i_1, \ldots, i_d) = p_1(i_1), (\forall i_1 \in [n])$$

$$\vdots \tag{3}$$

$$\sum_{i_1, \ldots, i_{d-1}} x(i_1, \ldots, i_d) = p_d(i_d), (\forall i_d \in [n]).$$

This linear program (LP) is central to the regularization schemes that are discussed below.

## 4 Efficient d-MMOT Computation

The linear d-MMOT problem (3) suffers from the curse of dimensionality: the LP has $n^d$ variables, and even modest choices of $n$ and $d$ can result in a LP with billions of variables, making standard LP solvers inapplicable. Alternatively, specific algorithms have been proposed (Benamou et al., 2015), and relaxations via entropic regularization have become more widespread, with very recent extensions to the d-MMOT setting (Tupitsa et al., 2020; Lin et al., 2022).

In practice, the cost $c$ in (3) typically takes one of two forms. In the case where the distributions $p_1, \ldots, p_d$ are over categorical variables, the cost is typically defined as $c(i_1, \ldots, i_d) = 0$ when $i_1 = \cdots = i_d$ and 1 otherwise. However, and importantly, if the distributions $p_i$ are histograms over some ordinal or discretized space, the cost typically has a structure closer to that of a "tensorized" distance, characterized by the *Monge* property.

**Definition 4.1** (Monge Property). A tensor $c$ is Monge if for all valid $i_1, \ldots i_d$ and $j_1, \ldots, j_d$,

$$c(s_1, \ldots, s_d) + c(t_1, \ldots t_d) \le c(i_1, \ldots i_d) + c(j_1, \ldots, j_d) \tag{4}$$

where $s_k = \min(i_k, j_k)$ and $t_k = \max(i_k, j_k)$.

Our focus is on a specific cost, which is known to be Monge: $c(i_1, i_2, \ldots, i_d) := \max\{i_k : k \in [d]\} - \min\{i_k : k \in [d]\}$. When $d = 2$, this cost reduces to $c(i_1, i_2) = |i_1 - i_2|$, which agrees with the classical EMD cost. This choice of $c$ is called the *generalized EMD cost*.

*Remark* 4.2. Limiting our attention to this cost is not as restrictive as it may appear. Indeed, Bein et al. (1995) shows that the optimal solution to the LP (3) is independent of the cost, as long as it is Monge. Additionally, when $c$ is the generalized EMD cost, Kline (2019) describes a greedy algorithm that solves both (3) and its dual (5) in linear time. It is also shown that the optimal objective value is a continuous nonnegative function of each probability distribution $p_j$ (i.e., small changes in one distribution cause small changes in the objective value). Continuity is critical for numerical stability. Next, when $c$ is the generalized Earth Mover's array, the optimal objective value vanishes if and only if $p_i = p_j$ for all $i, j \in [n]$. This *separability* property is useful in applications where we wish to iteratively "step towards" the barycenter of a set of distributions, see Fig. 2. In order to step towards (red arrows) a barycenter, we require a descent direction. We observe that the following result gives us this functionality.

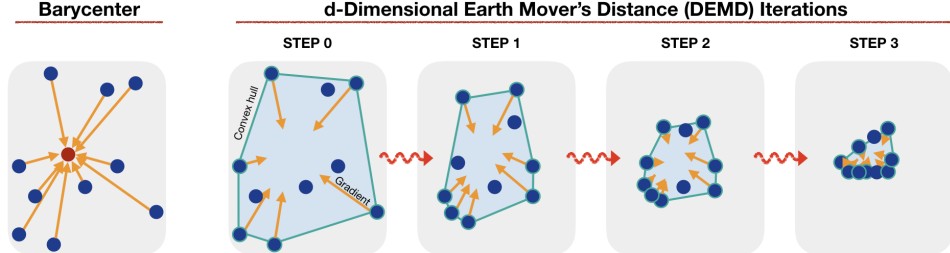

Figure 2: (*Left*) Barycenter methods identify a center (red circle) and transport *all* distributions (blue circles) toward that center along the coupling path (yellow). (*Right*) Our DEMD approach identifies "support" distributions that lie on the convex hull (outlined circles), and only those distributions are moved in a direction that decreases the Generalized EMD objective.

**Theorem 4.3.** *The dual linear program of the d-MMOT problem (3) is*

$$\underset{z_j \in \mathbb{R}^n, j \in [d]}{maximize} \qquad \sum_j p'_j z_j \qquad subject\ to \qquad z_1(i_1) + \cdots + z_d(i_d) \le c(i_1, \ldots, i_d), \qquad (5)$$

*where the indices in the constraints include all $i_j \in [n]$, $j \in [d]$. Denote by $\phi(p_1, \ldots, p_d)$, the optimal objective value of the LP in (3). Let $z^*$ be an optimal solution to the dual program (5). Then,*

$$\nabla \phi(p_1, \ldots, p_d) = z^*, \ and\ for\ any\ t \in \mathbb{R}, \ \phi(p_1, p_2, \ldots, p_d) = \sum_j p'_j(z_j^* + t\,\eta),$$

*where $\eta := (z_1^*(n)\,e, z_1^*(n)\,e, \cdots, z_d^*(n)\,e)$.*

*Proof.* The main observation invokes perturbation analysis (Mangasarian & Meyer, 1979; Ferris & Mangasarian, 1991) of LPs to assert that, under mild uniqueness conditions, small changes to a LP's input data does not change its optimal solution. The full proof is in Appendix 7.1. □

*Remark* 4.4. The first part of this result provides what we require: a direction of descent. Thus, if we can solve the d-MMOT problem and also find the optimal solution to its dual, then we can step (or move) our distributions in the opposite direction of the dual variables to push them together, see Fig. 2. The second claim is somewhat technical, and reconciles particular affine shifts that result in equivalent objective values.

### 4.1 OPTIMIZATION OF D-MMOT

**Setup.** We can now instantiate d-MMOT as an add-on term in a standard machine learning formulation. Concretely, it can be positioned alongside typical learning losses $L(f(x), y; \theta)$ to encourage minimizing distances among $d$ different distributions $g_i \in G, i \in [d]$, d-MMOT$(f(x), g; \theta)$, i.e., $\min_\theta L(f(x), y; \theta) + $ d-MMOT$(f(x), g; \theta)$. Within a deep neural network (DNN) architecture, as in some of our experiments, several properties of the d-MMOT module are useful: our results above naturally provide clean operations for computing both the required objective in the "forward" pass and gradients in the "backward" pass.

**Using the primal and dual variables.** If the optimal objective value of (3) serves in regularizing a deep neural network, then we can train the network as follows. During the forward pass, i.e., computing the d-MMOT objective, we can employ a version of the aforementioned primal/dual algorithm that solely computes the function $\phi$ and stores the dual variables $z$. As backpropagation proceeds, when the EMD module encounters an incoming gradient, it is simply multiplied by the stored dual variables (see Theorem 4.3 and Algorithm 1). We call our procedure the $d-$dimensional Earth Mover's Distance, or in short, *the DEMD algorithm.*

**Complexity.** Computing the DEMD distance in the forward pass is exactly $O(nd)$: linear in the number of distributions and number of bins. This property follows directly from the algorithm, needing only a single pass through all of the data. Bein et al. (1995) provides this result for greedy algorithms that solve OT programs as in Def. 3.1. In contrast to methods that derive gradients

---

**Algorithm 1** Our proposed method: $d-$Dimensional Earch Mover's Distance (DEMD)

---

    **function** FORWARD($p_1, \ldots, p_j$)
        Compute $\sum s_k t_k$ and $(z_1, \ldots, z_d)$ via the primal/dual algorithm in Kline (2019).
        save $(z_1, \ldots, z_d)$.
        **return** $\sum_k s_k t_k$
    **function** BACKWARD(gradOutput)
        load $(z_1, \ldots, z_d)$
        **return** $(z_1, \ldots, z_d) \cdot$ gradOutput

---

via entropic regularization schemes, i.e., relaxations of the optimal transport problem (Luise et al., 2018; Xie et al., 2020; Cuturi et al., 2020), this approach solves the distance computation exactly in linear time. This linear time analysis is not only provided by the theory in prior work, but is also explicit in the number of iterations defining our algorithm (see Appendix 7.4 for more details). For minimizing the DEMD (computing the updates, i.e., red arrows in Fig. 2), a convergence analysis would follow from the properties of the optimization scheme chosen. Our tool can be dropped in exactly as any other module in modern learning applications (using the observation that gradients are easily computed, i.e., $O(1)$ time to read stored dual variables).

**A few practical adjustments.** It often happens during training that the optimal solutions may, through updates by stepping in the direction of the gradient, acquire entries that are negative. This violates an assumption that entries must be nonnegative. However, Thm. 2.2 in Kline (2019) shows that the optimal objective value, $\phi$, possesses a type of translation invariance. Leveraging this alongside the second part of Thm 4.3, we ensure that, in the event that a point escapes the nonnegative orthant of $\mathbb{R}^n$, an appropriately constructed constant vector may be added to the current iterate so that it again lies in the nonnegative orthant, *without changing the objective value*. Further, with a relatively small learning rate, by the continuity of $\phi$, normalization is a small perturbation of the original point and can be applied as necessary to enforce that updates result in valid distributions.

*Remark* 4.5. Another useful property is that if the convex hull of $(p_1, \ldots, p_d)$ is contained in the convex hull of $(\hat{p}_1, \ldots, \hat{p}_{\hat{d}})$, then $\phi(p_1, \ldots, p_d) \leq \phi(\hat{p}_1, \ldots, \hat{p}_{\hat{d}})$, with strict inequality if containment is strict. A direct consequence of this property is that points within the interior of the convex hull of the data fed to the optimization model have vanishing gradients. Practically, this leads to **sparse** gradients w.r.t. the distributions, and nonzero gradients correspond to points on the hull, i.e., distributions which are maximally different from the rest. Minimization proceeds by iteratively moving mass such that these maximally different distributions are pushed toward each other. Fig. 2 shows the convex hull during minimization of the generalized EMD objective via our DEMD algorithm.

**Need for Histogramming.** Note that outputs $f(x)$ and intermediate activations are *continuous* values (layer shape by batch size). So, we must transform activations into normalized histograms (i.e., discrete distributions). While libraries such as PyTorch and Tensorflow typically provide histogram functions, they are not differentiable. The discontinuous operation of bin assignment does not allow for an end-to-end pipeline where we can push upstream parameters in the direction of minimizing the EMD objective over histograms. To address this, we use a simple relaxed/differentiable histogramming operation. First, all outputs are mapped to $[0, 1]$ using a Sigmoid activation. Then, for each bin location, the "count" in that bin is determined by a ReLU function applied to the difference between the gap between the activation and the distance to the bin boundary. In other words, if an activation falls in a bin, the count of that bin increases based on the distance to the bin boundary, otherwise the count remains the same. The full procedure is detailed in Appendix 7.2. The ReLU activations defining bin boundaries allow for gradients to move samples towards neighboring bins as needed.

## 5 EXPERIMENTS

We evaluate our construction in a number of settings. First, we demonstrate the computational speedup associated with evaluating d-MMOT using our algorithm, along with speedups associated with directly computing the gradient. Next, we compute the construction in a series of neural network tasks associated with ensuring distributional similarity: fairness, invariant representations, and multi-domain matching. We provide a complete PyTorch Network Module that packages the above

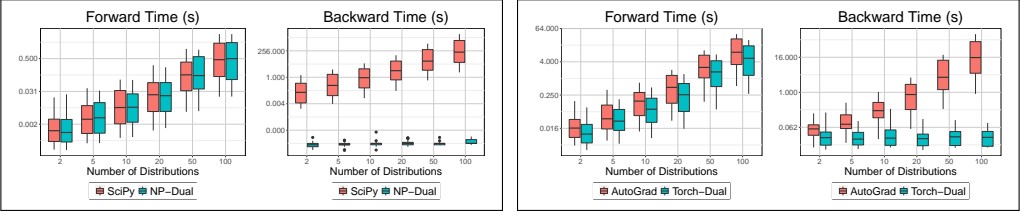

Figure 3: (Left) DEMD is positioned after the final layer, prior to the activation. Activations are sorted into distributions utilizing group labels provided alongside the input data. Computed distributions are then brought together using our algorithm. (Right) Computation times for direct distance evaluation of EMD-like distances. Existing methods take much more time ($y$-axis is exponential) as the number of distributions grow.

differentiable DEMD objective and histogram functions, and serves as a plug-and-play regularization module. Code is available at https://github.com/ronakrm/demd.

## 5.1 PERFORMANCE BENCHMARKS

Figure 4 presents NumPy and Torch instantiations of both forward and backward passes using automatic differentiation and the gradients computed using the dual as in §4.1. As expected, the forward (distance computation) times are comparable, but the backwards computation scales poorly with the number of distributions to be updated using automatic differentiation. Our dual setting allows the gradients to simply be read off (based on the forward pass), leading to **no** additional computation overhead during backpropagation.

Figure 4: Forward/Backward Pass Times for 10 Bins with varying number of distributions, averaged over 10 runs. Forward wall-clock times are comparable, regardless of backend (*left pair*: NumPy+SciPy; *right pair*: PyTorch+AutoGrad). Direct reading of the gradient via the dual leads to significant gains in backward pass times, where automatic differentiation scales poorly with the number of distributions.

We compare our DEMD computation to what one may use given existing Optimal Transport methods in Figure 3 (right). Using standard off-the-shelf methods as a baseline, when the cost is Monge our algorithm provides *significant* speedups in computation time, on the scale of orders of magnitude! Further, if the number of distributions and bins increases, the time-cost for existing methods using more generic LP solvers can become significant, and may become infeasible with generic solutions via CVXPY (Diamond & Boyd, 2016). We compare our approach to two off-the-shelf implementations of barycenters via the Python Optimal Transport (POT) Library (Flamary et al., 2021), along with directly solving the Earth Mover's problem via CVXPY. Even for 10 distributions over 10 bins, CVXPY is unable to allocate the necessary memory using a direct instantiation.

## 5.2 GENERALIZED EM FAIRNESS ON FAIRNESS DATASETS

With a viable tool in hand, we move to practical applications in machine learning fairness, which naturally requires enforcing closeness in model outputs. Here, we construct networks with our DEMD regularizer, where we discretize the final activation output prior to classification, and push the distributions of this activation to be similar among sensitive attributes.

**Data.** We identify 4 common fairness datasets often used to benchmark fair machine learning algorithms: (1) the German Credit Dataset (Hofmann, 1994), (2) the Adult Income Dataset (Dua & Graff, 2017), (3) the Communities and Crime Dataset (Redmond, 2009), and (4) The ACS Income dataset, recently made available as a large, population-level demographic dataset (Ding et al., 2021)

Table 1: **Fairness Experiments.** Measures evaluated using standard metrics: maximum Demographic Parity Gap (**DP**), maximum Equalized Odds Gap (**EO**), and (**DEMD**). For all measures, lower values are preferred. With comparable accuracy, DEMD regularization leads to fairer representations as measured by common metrics. DP and EO measures are scaled by 100 for ease of presentation. Best results shown in bold.

| | German | | | Adult | | | Crime | | | ACS-Income | | |
|---|---|---|---|---|---|---|---|---|---|---|---|---|
| | DP | EO | DEMD | DP | EO | DEMD | DP | EO | DEMD | DP | EO | DEMD |
| None | $17_{(5)}$ | $11_{(2)}$ | $1.69_{(0.32)}$ | $18_{(1)}$ | $13_{(0)}$ | $1.69_{(0.07)}$ | $38_{(6)}$ | $45_{(3)}$ | $2.86_{(0.38)}$ | $37_{(1)}$ | $25_{(0)}$ | $4.78_{(0.32)}$ |
| DP-Reg. | $16_{(6)}$ | $10_{(3)}$ | $1.5_{(0.26)}$ | $17_{(1)}$ | $13_{(1)}$ | $1.6_{(0.07)}$ | $38_{(6)}$ | $45_{(3)}$ | $2.83_{(0.39)}$ | $48_{(4)}$ | $28_{(0)}$ | $5.02_{(0.31)}$ |
| EO-Reg. | $17_{(5)}$ | $11_{(2)}$ | $1.69_{(0.32)}$ | $\mathbf{14}_{(1)}$ | $12_{(1)}$ | $\mathbf{1.43}_{(0.07)}$ | $38_{(5)}$ | $\mathbf{44}_{(3)}$ | $2.83_{(0.39)}$ | $38_{(1)}$ | $26_{(0)}$ | $4.82_{(0.32)}$ |
| Bary-POT | $27_{(5)}$ | $17_{(1)}$ | $1.5_{(0.21)}$ | $18_{(1)}$ | $13_{(0)}$ | $1.64_{(0.07)}$ | $\mathbf{36}_{(5)}$ | $\mathbf{44}_{(4)}$ | $2.81_{(0.3)}$ | $57_{(37)}$ | $50_{(44)}$ | $4.38_{(0.16)}$ |
| DEMD (ours) | $\mathbf{14}_{(7)}$ | $\mathbf{9}_{(4)}$ | $\mathbf{1.41}_{(0.35)}$ | $15_{(1)}$ | $\mathbf{12}_{(1)}$ | $1.44_{(0.08)}$ | $\mathbf{36}_{(6)}$ | $\mathbf{44}_{(3)}$ | $\mathbf{2.69}_{(0.44)}$ | $\mathbf{33}_{(0)}$ | $\mathbf{24}_{(0)}$ | $\mathbf{3.6}_{(0.29)}$ |

containing as many as nine sensitive attributes. We set up a simple three-layer neural network for classification tasks with the addition of a fairness-type regularizer. We compare our construction with 4 off-the-shelf plug-in regularizers: (1) No regularization, (2) Demographic Parity (DP), (3) Equalized Odds (EO), and (4) a histogrammed barycenter construction. DP and EO regularizers were computed using a PyTorch version of FairLearn (Bird et al., 2020), and the barycenter version was implemented using POT library (with GPU backend). Because the scale of the regularization term is not directly comparable, we sweep regularization weights and select the best over all measures for a each dataset/regularizer pair. We use 10 bins and replicate all experiments over three random seeds.

**Models.** We set up two model settings with a standard logistic regressor and a 2-layer neural network. We compare three types of plug-in regularizers: (1) Demographic Parity (DP), (2) Equalized Odds (EO), and (3) the Generalized EMD. DP and EO regularizers were computed using a PyTorch implementation of FairLearn (Bird et al., 2020).

**Results.** In the summary in Table 1, models were selected with the largest regularization weight before accuracy dropped significantly. When accuracies are comparable, we see good performance (when minimizing DEMD regularizer) against baseline methods. Notably, when accuracies are similar across methods, minimizing DEMD tends to give better (lower) fairness measures across all datasets. **Summary:** DEMD on the final network layer helps control multiple fairness measures.

## 5.3 HARMONIZATION FOR INVARIANT REPRESENTATIONS

Having evaluated the use of the DEMD layer to constrain the neural network for fairness measures, we will now move to a more general problem of deriving invariant representations from the datasets. Here, invariance is sought w.r.t. the sensitive attributes. Recent works (Lokhande et al., 2022a; Akash et al., 2021) on invariant representation learning propose leveraging an encoder-decoder architectures to map the dataset features to latent space representations. The latent space representations are penalized to match or harmonize the distributions across several groups in the dataset. In contrast to the previous section, the goal here is to identify a good mapping in the latent space that is devoid of any group-related information in the dataset. Consequently, our evaluation metrics test if the latent representations have a lower value of (i) ADV, adversarial measure and (ii) MMD, maximum mean discrepancy measure. The measure ADV tests to what extent can a separate neural network predict the group information from the latent features. Alternatively, the MMD scores measure the distance between the probability distributions across groups. Prior works such as Li et al. (2014) and Xie et al. (2017) optimize each of these measures separately. Our experiments (Table 2) show the DEMD layer performs competitively with the baselines when applied on the latent space. Interestingly, these performance gains come despite not directly optimizing the harmonization measures, in contrast to baselines, which require several practical adjustments (batch variants and secondary neural networks). **Summary:** DEMD as an intermediate layer can be used to derive invariant representations efficiently.

## 5.4 MULTI-DOMAIN IMAGE TRANSLATION

We apply our construction to a recent multi-marginal GAN framing of multi-domain image translation. With the goal of learning a mapping for a source image to multiple target domains, a focal point of recent literature has been to reduce the computational requirements of training individual models for each target, and learn a concurrent matching problem over a number of shared parameters or networks. MWGAN (Cao et al., 2019) set up a Multiple Marginal Matching problem in this context. Extending upon their key observation that inter-domain constraints can be measured via the gradients

Table 2: **Harmonization Experiments.** Evaluations conducted along three metrics, Accuracy (**ACC**), Adversarial measure (**ADV**) and Maximum Mean Discrepancy (**MMD**). A lower value ↓ of ADV and MMD indicate successful harmonization across the different groups. A higher ACC with a small drop from baseline is preferred.

| | German | | | Adult | | | Crime | | | ACS-Income | | |
|---|---|---|---|---|---|---|---|---|---|---|---|---|
| | ACC ↑ | ADV ↓ | MMD ↓ | ACC ↑ | ADV ↓ | MMD ↓ | ACC ↑ | ADV ↓ | MMD ↓ | ACC ↑ | ADV ↓ | MMD ↓ |
| None | $74_{(0.9)}$ | $93_{(1.3)}$ | $7.7_{(0.8)}$ | $84_{(0.1)}$ | $83_{(0.1)}$ | $9.8_{(0.3)}$ | $85_{(0.2)}$ | $77_{(0.5)}$ | $14_{(0.1)}$ | $78_{(0.1)}$ | $98_{(0.7)}$ | $160_{(2)}$ |
| Li et al. (2014) | $73_{(1.5)}$ | $92_{(1.3)}$ | $1.5_{(0.3)}$ | $84_{(0.1)}$ | $83_{(0.1)}$ | $3.1_{(0.3)}$ | $85_{(0.5)}$ | $76_{(1.6)}$ | $12_{(1.0)}$ | $78_{(0.1)}$ | $97_{(0.5)}$ | $17_{(1.2)}$ |
| Xie et al. (2017) | $76_{(1.3)}$ | $93_{(0.6)}$ | $1.2_{(0.2)}$ | $84_{(0.04)}$ | $81_{(0.7)}$ | $4.2_{(2.4)}$ | $85_{(0.2)}$ | $76_{(0.7)}$ | $15_{(0.6)}$ | $78_{(0.1)}$ | $94_{(5.6)}$ | $99_{(2.9)}$ |
| DEMD (ours) | $74_{(1.1)}$ | $93_{(0.4)}$ | $2.1_{(0.4)}$ | $84_{(0.03)}$ | $82_{(0.2)}$ | $5.3_{(1.4)}$ | $83_{(0.3)}$ | $72_{(1.0)}$ | $7.1_{(1.0)}$ | $77_{(0.4)}$ | $96_{(0.5)}$ | $26_{(3.9)}$ |

| Original | Blond Hair | Glasses | Moustache | Pale Skin | Original | Blond Hair | Glasses | Moustache | Pale Skin |

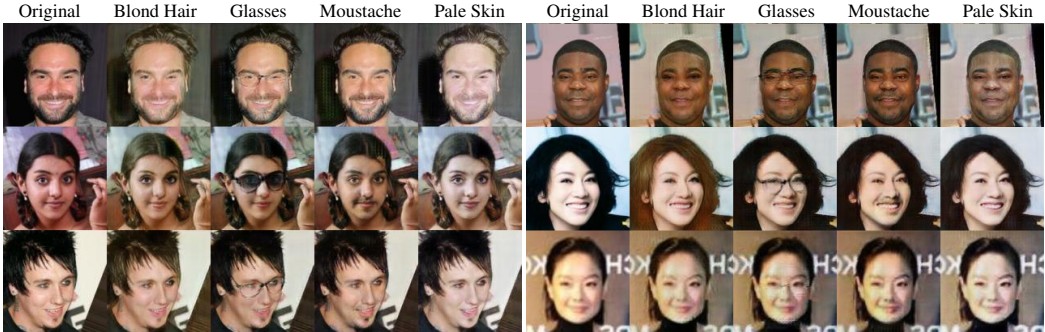

Figure 5: Six MWGAN+DEMD CelebA image translation results. Each row corresponds to a random sample in the validation set. The leftmost image is the original source image, and sequential columns represent translations to (1) Blonde Hair, (2) Glasses, (3) Moustache, and (4) Pale Skin. Using DEMD as a relaxation to the Multiple Marginal Matching problem facilitates high quality images translation. For more results see Appendix 7.3.4.

for each domain, we instantiate our DEMD layer over the gradient norms computed for each sample in a batch per group. Specifically, DEMD minimizes the differences between the distributions over the gradient norms across all target domains. Using a weight of 100 for both regularizers, we observe similar performance when compared to the original MWGAN construction. In Figure 5 we show a few samples generated over an image translation task on the CelebA dataset. Here, we aim to translate original dataset images to ones with specific target attributes. **Summary:** Multi-domain image translation can be conducted by adding a DEMD layer in the native GAN construction.

# 6 DISCUSSION

We presented an efficient solution for solving common practical multi-marginal optimal transport problems. Significantly cheaper to compute compared to similar methods, it allows for large numbers of distributions to be matched in common DNN pipelines. Our implementation allows imposing fairness constraints for a variety of applications, without the need for pairwise measures. As such, subgroup fairness (Kearns et al., 2018) is an interesting problem setting that we believe can benefit. Other properties such as Minkowski additivity that have not been explicitly leveraged in our experiments may also be a worthwhile direction to explore.

In its current form, DEMD cannot be directly applied to distributions over multi-dimensional discrete spaces, such as images or latent spaces in generative models. Slicing is a heuristic that has been shown to work well. To evaluate feasibility, we embed distributions over multi-dimensional continuous spaces, take random projections over 1-D spaces, and recompute our DEMD measure. Over a 64-dimensional latent space embedding of CelebA, we can efficiently compute our DEMD measure over all 40 attribute subgroups, and observe convergent behavior w.r.t. the number of projections.

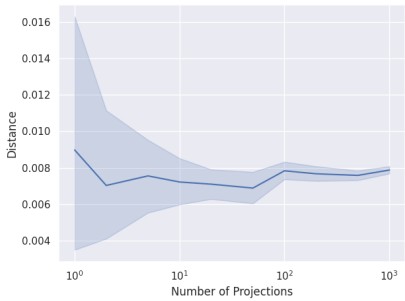

Figure 6: Sliced DEMD across projections.

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

## 7 Appendix

### 7.1 Proof of Theorem 4.2

Here, we present the proof of Theorem 4.2. The main observation in the proof invokes perturbation analysis Mangasarian & Meyer (1979); Ferris & Mangasarian (1991) of linear programs to assert that, under mild uniqueness conditions, small changes to a linear program's data does not change the linear program's optimal solution. The informal reason why such a result is possible can be explained using a short, geometric argument as follows.

The feasible set of a nontrivial linear program is a polytope, and, as a rule, an optimal solution to a linear program lives at the point where a hyperplane defined by the objective functional intersects a vertex of the polytope. A small perturbation of the hyperplane does not alter the intersecting vertex. The derivative of a linear program's optimal objective value, treated as a function of the data fed to the LP, has been described several times in prior work, and under a variety of conditions De Wolf & Smeers (2019); Freund (1985); Agueh & Carlier (2011); Mills (1957).

The second claim of Theorem 4.2 is useful because dual solutions to the generalized EMD linear program are not unique. The claim explains how one can modify a given solution (found by a direct gradient computation or else from an interior point solver, for example) so that it agrees with the solution yielded by the primal/dual greedy algorithm.

*Proof of Theorem 4.2.* Let $z_j^*$, for $j \in [d]$, denote an optimal solution to the dual linear program of equation equation 5 in the main paper. Standard sensitivity analysis of linear programs implies that $z_j^* \in \mathbb{R}^n$ ($j \in [d]$) is also optimal for the perturbed linear program,

$$\underset{z_j \in \mathbb{R}^n, j \in [d]}{\text{maximize}} \qquad \sum_j (x_j + \varepsilon h_j)' z_j$$

$$\text{subject to} \qquad z_1(i_1) + \cdots + z_d(i_d) \leq c(i_1, \ldots, i_d),$$

where the indices in the constraints include all $i_j \in [n]$, $j \in [d]$, $\varepsilon > 0$ is sufficiently small and $h_j \in \mathbb{R}^n$ are held fixed.

If $\phi(x_1 + \varepsilon h_1, \ldots, x_d + \varepsilon h_d)$ represents the optimal objective value of this program, then by linearity,

$$\phi(x_1 + \varepsilon h_1, \ldots, x_d + \varepsilon h_d) - \phi(x_1, \ldots, x_d) = \sum_j h_j' z_j^*.$$

Thus, we can form the directional derivative of $\phi$ as

$$\lim_{\varepsilon \to 0} \frac{\phi(x_1 + \varepsilon h_1, \ldots, x_d + \varepsilon h_d) - \phi(x_1, \ldots, x_d)}{\varepsilon \|h\|} = \frac{\sum_j h_j' z_j^*}{\|h\|},$$

where $\|h\|^2 := \sum_j \|h_j\|^2$. From this, it follows that

$$\nabla \phi(x_1, \ldots, x_d) = (z_1^*, z_2^*, \ldots, z_d^*).$$

This shows the first claim of the theorem.

To see the second claim, we have from Theorem 3.1 item 2 of Kline (2019) that

$$\sum_j z_j^*(i) = 0 \tag{6}$$

for all $i \in [n]$. Consequently, if one defines

$$\eta = (z_1^*(n)e, z_2^*(n)e, \cdots, z_d^*(n)e),$$

then since we assume that $e'x_j = 1$ for all $j \in [d]$,

$$\sum_j x_j'(z_j^* + t\eta) = \sum_j x_j' z_j^* + t\, x_j' \eta$$

$$= \sum_j x_j' z_j^* + t \sum_j z_j^*(n)\, x_j' e$$

$$= \sum_j x_j' z_j^*,$$

where the last equality holds by equation (6). This shows the second claim. $\square$

## 7.2 Differentiable Histogramming

---
**Algorithm 2** Differentiable Histograms

---
1: **function** INIT($n$) *// bins, (discretization)*
2: $\quad r := 1/n$ *// bin size*
3: $\quad locs := arange(0, 1, r)$ *// bin boundaries*
4: **function** FORWARD($acts$)
5: $\quad cdfs = \sigma(acts)$ *// compute CDFs*
6: $\quad counts = []$
7: $\quad$ **for** loc in locs **do**
8: $\quad\quad dist = |cdfs - loc|$ *// dist. to boundary*
9: $\quad\quad ct = \sum_{i \in [nbins]} \text{ReLU}(r - dist[i])$ *// soft bucket count*
10: $\quad\quad counts.append(ct)$
11: $\quad out = stack(counts)$
12: $\quad out = out/sum(out)$
13: $\quad$ **return** $out$

---

While gradients are now readily available, typical ML pipelines do not have distributions or histograms predefined at outputs which can be fed directly into our EMD loss. Applying existing binning procedures over the batch to estimate histograms will break the ability to autodifferentiate: soft thresholds are necessary at bin boundaries such that samples within a bin may move smoothly as needed. Algorithm 2 provides a differentiable histogram implementation. Using a rectified linear relaxation allows for samples to have a continuous gradient towards neighboring bins.

## 7.3 Experimental Details

Results reported in tables in the main paper are of the form $M_{(SD)}$, where $M$ is the mean and $SD$ is the standard deviation calculated over replications.

**Setup details.** Experiments were conducted using NumPy and PyTorch on a Intel(R) Xeon(R) CPU E5-2620 v3 @ 2.40GHz with an Nvidia Titan Xp GPU. Particular parameter settings and experimental runs can be found below or in the scripts included with the code provided at `https://github.com/ronakrm/demd`.

### 7.3.1 Data and Licenses

The datasets Adult, Communities and Crime, and German datasets are all available under Creative Commons Attribution 4.0 International (CC BY 4.0) licenses via the UCI Machine Learning Dataset Repository `https://archive.ics.uci.edu/ml/index.php`. The CelebA dataset `http://mmlab.ie.cuhk.edu.hk/projects/CelebA.html` is available for non-commercial purposes. See the website for more details.

**ACS Data.** The American Census Survey (ACS) has recently made available a large set of demographic data. The original UCI Adult dataset Dua & Graff (2017) was curated from this data, however recent work by Ding et al. (2021) has identified temporal shifts in demographic data, and recommends using a more recent collection as a baseline when evaluating biases and adjusting for fairness. Part of their contribution includes APIs to directly interface with the data provided by the ACS, and the ability to identify and construct similar problems associated with the original UCI-provided dataset, albeit with updated data. Data for the income prediction task was downloaded from 2018, localized to Louisiana. Race is the provided group label, which we wish to be agnostic towards, over some measure of our output. Data was accessed using the folktables codebase `https://github.com/zykls/folktables` with MIT License. The US Census data accessed is available for use so long as it is not used in combination with other data "to identify any particular respondent to a Census Bureau survey." See `https://www.census.gov/data/developers/about/terms-of-service.html` for more details.

**German Data.** The German dataset classifies people as good or bad credit risks. There are about 20 features (7 numerical and 13 categorical). These features represent the economic status of the person, such as, credit history, savings account, year of present employment, property and others.

**Adult Data.** The Adult dataset is comprised of demographic characteristics from the UCI repository Dua & Graff (2017). The protected attribute here is gender. It contains $44,842$ samples. The features that were used in the experiments include "age", "workclass", "fnlwgt", "education", "education-num", "marital-status", "occupation", "relationship", "race", "sex", "capital-gain", "capital-loss", "hours-per-week", "native-country", "income". A positve target label in this dataset is indicated by the attribute "income-bracket" being above $50K.

**Communities and Crime Data.** The Communities and Crime dataset consists of summary statistics of per-capita measures from a wide variety of communities across the United States, measured from a number of US census and surveys from the 1990's. The original goal of the dataset was to predict crime rates in communities as a function of various demographic, socioeconomic, and other features. The dataset has widely become known as the prototypical example in which using racial population distributions can be extremely harmful in perpetuating stereotypes and lead to models that continue to exacerbate inequity that may exist within the data. As such, the dataset has become a de facto tool in evaluating fairness metrics and methods that attempt to account for these inequities and biases. The preprocessed data contains 1994 samples, and we attempt to predict the violent crime rate (binarized at 0.3 after normalization between 0 and 1), and the sensitive attribute is a similarly binarized version of the percentage black population variable.

**CelebA Data.** CelebA (Liu et al., 2015) consists of 200K celebrity face images from the internet annotated by a group of paid adult participants. There are up to $40$ labels available in the dataset, each of which is binary-valued.

### 7.3.2 FAIRNESS EXPERIMENT DETAILS

All experiments related to fairness use a three-layer fully-connected neural network classifier with a hidden layer size of 100. Numbers reported in Table 1 in the main paper are means and standard deviations over three replicate runs with different seeds. Hyperparameters were selected as described in the main paper, by taking the best result for each dataset over the parameter range $\lambda \in [1.0, 0.1, 10, 0.01, 100, 0.001]$. The full results of this sweep are presented in Table 3 and Table 4.

The $|DP|$ and $|EO|$ measures are the gap between the corresponding conditional probability distributions when there is a single binary group attribute, and the largest gap ($max - min$) when there are more than 2 groups. The DEMD loss is computed as described in the main paper with a discretization/bin level of 10.

### 7.3.3 HARMONIZATION EXPERIMENT DETAILS

For all our experiments related to harmonization, we use an encoder-decoder framework comprising of fully-connected layers. The hidden layers comprised of $64$ nodes and the latent space was of dimension $30$. We report the mean and standard deviation on unseen test datasets for three random runs. The hyper-parameter selection has been done on a validation split obtained from the training dataset. The model that achieves the best ADV (the adversarial evaluation measure), with the test set accuracy remains within 5% of the vanilla (titled "None" in the paper) model, is chosen.

Recall that the harmonization experiments aimed at minimizing the distributional differences of the latent features across the groups. Below, we will provide details on the evaluation metrics, ADV and MMD measures, that aptly assess distributional differences of the latent features.

**Evaluation Metric: Adversarial Measure (ADV)** The ADV measure corresponds to the accuracy obtained by training a separate neural network to predict groups from the latent features. This step is conducted post harmonization. A lower value of the ADV accuracy denotes that the latent features are free of any group related information. This implies successful harmonization and suggests minimal distributional differences of the latent features across groups. We follow (Xie et al., 2017) for training the adversary used for reporting ADV measure. We use a three-layered fully-connected network with batch normalization and train it with Adam optimizer for 200 epochs. The learning rate for the adversary is decreased multiplicatively by a factor of $0.65$ every 10 epochs for convergence.

**Evaluation Metric: Maximum Mean Discrepancy (MMD)** We simply use the following MMD criterion as described in (Gretton et al., 2006) and evaluate the metric on the latent features obtained from the test set. Each group is considered as a different distribution and a lower value of this metric

Table 3: **Fairness Experiments.** Measures evaluated using standard metrics: maximum Demographic Parity Gap (**—DP—**), maximum Equalized Odds Gap (**—EO—**), and (**DEMD**). For all measures, lower values are preferred. With comparable accuracy, DEMD regularization leads to fairer representations as measured by common metrics. DP and EO measures are scaled by 100 for ease of presentation. Best results shown in bold.

| Reg. Type | $\lambda$ | German | | | Adult | | |
|---|---|---|---|---|---|---|---|
| | | —DP— | —EO— | DEMD | —DP— | —EO— | DEMD |
| None | 0 | $0.17_{(0.05)}$ | $0.11_{(0.02)}$ | $1.69_{(0.32)}$ | $0.18_{(0.01)}$ | $0.13_{(0.01)}$ | $1.69_{(0.07)}$ |
| DP-Reg. | 0.001 | $0.19_{(0.03)}$ | $0.12_{(0.0)}$ | $1.79_{(0.27)}$ | $0.18_{(0.01)}$ | $0.13_{(0.0)}$ | $1.64_{(0.07)}$ |
| | 0.01 | $0.27_{(0.05)}$ | $0.17_{(0.01)}$ | $1.5_{(0.21)}$ | $0.13_{(0.01)}$ | $0.14_{(0.01)}$ | $0.99_{(0.09)}$ |
| | 0.1 | $0.23_{(0.2)}$ | $0.12_{(0.1)}$ | $0.99_{(0.86)}$ | $0.02_{(0.03)}$ | $0.15_{(0.02)}$ | $0.28_{(0.09)}$ |
| | 1.0 | $0.0_{(0.0)}$ | $0.19_{(0.17)}$ | $0.0_{(0.0)}$ | $0.0_{(0.0)}$ | $0.2_{(0.0)}$ | $0.0_{(0.0)}$ |
| | 10.0 | $0.0_{(0.0)}$ | $0.1_{(0.17)}$ | $0.0_{(0.0)}$ | $0.0_{(0.0)}$ | $0.2_{(0.0)}$ | $0.0_{(0.0)}$ |
| | 100.0 | $0.0_{(0.0)}$ | $0.0_{(0.0)}$ | $0.0_{(0.0)}$ | $0.0_{(0.0)}$ | $0.2_{(0.0)}$ | $0.0_{(0.0)}$ |
| EO-Reg. | 0.001 | $0.17_{(0.05)}$ | $0.1_{(0.02)}$ | $1.51_{(0.34)}$ | $0.18_{(0.01)}$ | $0.13_{(0.01)}$ | $1.67_{(0.07)}$ |
| | 0.01 | $0.14_{(0.07)}$ | $0.09_{(0.04)}$ | $1.41_{(0.35)}$ | $0.15_{(0.01)}$ | $0.12_{(0.01)}$ | $1.44_{(0.08)}$ |
| | 0.1 | $0.0_{(0.0)}$ | $0.0_{(0.0)}$ | $0.47_{(0.21)}$ | $0.03_{(0.01)}$ | $0.06_{(0.01)}$ | $0.29_{(0.02)}$ |
| | 1.0 | $0.0_{(0.0)}$ | $0.0_{(0.0)}$ | $0.08_{(0.14)}$ | $0.0_{(0.0)}$ | $0.0_{(0.0)}$ | $0.0_{(0.0)}$ |
| | 10.0 | $0.0_{(0.0)}$ | $0.19_{(0.17)}$ | $0.0_{(0.0)}$ | $0.0_{(0.0)}$ | $0.0_{(0.0)}$ | $0.0_{(0.0)}$ |
| | 100.0 | $0.0_{(0.0)}$ | $0.0_{(0.0)}$ | $0.0_{(0.0)}$ | $0.0_{(0.0)}$ | $0.2_{(0.0)}$ | $0.0_{(0.0)}$ |
| Bary | 0.001 | $0.17_{(0.05)}$ | $0.11_{(0.02)}$ | $1.69_{(0.32)}$ | $0.18_{(0.01)}$ | $0.13_{(0.01)}$ | $1.69_{(0.07)}$ |
| | 0.01 | $0.17_{(0.05)}$ | $0.11_{(0.02)}$ | $1.68_{(0.31)}$ | $0.18_{(0.01)}$ | $0.13_{(0.01)}$ | $1.69_{(0.07)}$ |
| | 0.1 | $0.17_{(0.05)}$ | $0.1_{(0.02)}$ | $1.51_{(0.35)}$ | $0.18_{(0.01)}$ | $0.13_{(0.01)}$ | $1.68_{(0.07)}$ |
| | 1.0 | $0.16_{(0.06)}$ | $0.1_{(0.03)}$ | $1.5_{(0.26)}$ | $0.17_{(0.01)}$ | $0.13_{(0.01)}$ | $1.6_{(0.07)}$ |
| | 10.0 | $0.06_{(0.07)}$ | $0.04_{(0.05)}$ | $0.58_{(0.37)}$ | $0.09_{(0.01)}$ | $0.09_{(0.01)}$ | $1.02_{(0.08)}$ |
| | 100.0 | $0.06_{(0.11)}$ | $0.03_{(0.06)}$ | $0.06_{(0.11)}$ | $0.0_{(0.0)}$ | $0.04_{(0.0)}$ | $0.39_{(0.05)}$ |
| DEMD | 0.001 | $0.17_{(0.05)}$ | $0.11_{(0.02)}$ | $1.69_{(0.32)}$ | $0.18_{(0.01)}$ | $0.13_{(0.01)}$ | $1.69_{(0.07)}$ |
| | 0.01 | $0.17_{(0.05)}$ | $0.11_{(0.02)}$ | $1.69_{(0.32)}$ | $0.18_{(0.01)}$ | $0.13_{(0.01)}$ | $1.69_{(0.07)}$ |
| | 0.1 | $0.19_{(0.05)}$ | $0.11_{(0.02)}$ | $1.32_{(0.18)}$ | $0.18_{(0.01)}$ | $0.13_{(0.01)}$ | $1.66_{(0.07)}$ |
| | 1.0 | $0.2_{(0.09)}$ | $0.11_{(0.05)}$ | $1.59_{(0.46)}$ | $0.14_{(0.01)}$ | $0.12_{(0.01)}$ | $1.43_{(0.07)}$ |
| | 10.0 | $0.16_{(0.13)}$ | $0.09_{(0.07)}$ | $0.5_{(0.12)}$ | $0.07_{(0.02)}$ | $0.08_{(0.01)}$ | $0.94_{(0.17)}$ |
| | 100.0 | $0.06_{(0.08)}$ | $0.18_{(0.16)}$ | $0.06_{(0.08)}$ | $0.01_{(0.01)}$ | $0.01_{(0.01)}$ | $0.41_{(0.05)}$ |

Table 4: **Fairness Experiments.** Measures evaluated using standard metrics: maximum Demographic Parity Gap (**—DP—**), maximum Equalized Odds Gap (**—EO—**), and (**DEMD**). For all measures, lower values are preferred. With comparable accuracy, DEMD regularization leads to fairer representations as measured by common metrics. DP and EO measures are scaled by 100 for ease of presentation. Best results shown in bold.

| Reg. Type | $\lambda$ | Crime | | | ACS-Income | | |
|---|---|---|---|---|---|---|---|
| | | —DP— | —EO— | DEMD | —DP— | —EO— | DEMD |
| None | 0 | $0.38_{(0.06)}$ | $0.45_{(0.03)}$ | $2.86_{(0.38)}$ | $0.37_{(0.01)}$ | $0.25_{(0.0)}$ | $4.78_{(0.32)}$ |
| DP-Reg. | 0.001 | $0.36_{(0.05)}$ | $0.44_{(0.04)}$ | $2.81_{(0.3)}$ | $0.57_{(0.37)}$ | $0.5_{(0.44)}$ | $4.38_{(0.16)}$ |
| | 0.01 | $0.35_{(0.02)}$ | $0.44_{(0.03)}$ | $2.61_{(0.17)}$ | $0.81_{(0.33)}$ | $0.76_{(0.42)}$ | $3.45_{(0.71)}$ |
| | 0.1 | $0.22_{(0.07)}$ | $0.29_{(0.12)}$ | $1.58_{(0.28)}$ | $0.62_{(0.33)}$ | $0.51_{(0.42)}$ | $2.67_{(0.63)}$ |
| | 1.0 | $0.0_{(0.0)}$ | $0.41_{(0.03)}$ | $0.0_{(0.0)}$ | $0.0_{(0.0)}$ | $0.67_{(0.58)}$ | $0.0_{(0.0)}$ |
| | 10.0 | $0.0_{(0.0)}$ | $0.26_{(0.23)}$ | $0.0_{(0.0)}$ | $0.0_{(0.0)}$ | $0.33_{(0.58)}$ | $0.0_{(0.0)}$ |
| | 100.0 | $0.0_{(0.0)}$ | $0.26_{(0.23)}$ | $0.0_{(0.0)}$ | $0.0_{(0.0)}$ | $0.33_{(0.58)}$ | $0.0_{(0.0)}$ |
| EO-Reg. | 0.001 | $0.38_{(0.05)}$ | $0.45_{(0.03)}$ | $2.85_{(0.38)}$ | $0.36_{(0.01)}$ | $0.25_{(0.0)}$ | $4.77_{(0.32)}$ |
| | 0.01 | $0.36_{(0.06)}$ | $0.44_{(0.03)}$ | $2.69_{(0.44)}$ | $0.33_{(0.0)}$ | $0.24_{(0.0)}$ | $3.6_{(0.29)}$ |
| | 0.1 | $0.11_{(0.07)}$ | $0.39_{(0.03)}$ | $0.61_{(0.23)}$ | $0.03_{(0.0)}$ | $0.03_{(0.0)}$ | $0.53_{(0.01)}$ |
| | 1.0 | $0.0_{(0.0)}$ | $0.41_{(0.03)}$ | $0.0_{(0.0)}$ | $0.0_{(0.0)}$ | $0.0_{(0.0)}$ | $0.0_{(0.0)}$ |
| | 10.0 | $0.0_{(0.0)}$ | $0.41_{(0.03)}$ | $0.0_{(0.0)}$ | $0.0_{(0.0)}$ | $0.33_{(0.58)}$ | $0.0_{(0.0)}$ |
| | 100.0 | $0.06_{(0.1)}$ | $0.09_{(0.15)}$ | $0.5_{(0.87)}$ | $0.0_{(0.0)}$ | $0.33_{(0.58)}$ | $0.0_{(0.0)}$ |
| Bary | 0.001 | $0.38_{(0.05)}$ | $0.45_{(0.03)}$ | $2.86_{(0.38)}$ | $0.37_{(0.01)}$ | $0.25_{(0.0)}$ | $4.78_{(0.32)}$ |
| | 0.01 | $0.38_{(0.06)}$ | $0.45_{(0.03)}$ | $2.86_{(0.39)}$ | $0.37_{(0.01)}$ | $0.25_{(0.0)}$ | $4.8_{(0.32)}$ |
| | 0.1 | $0.38_{(0.06)}$ | $0.45_{(0.03)}$ | $2.83_{(0.39)}$ | $0.48_{(0.04)}$ | $0.28_{(0.01)}$ | $5.02_{(0.31)}$ |
| | 1.0 | $0.37_{(0.06)}$ | $0.44_{(0.03)}$ | $2.65_{(0.45)}$ | $1.0_{(0.0)}$ | $1.0_{(0.0)}$ | $5.66_{(0.3)}$ |
| | 10.0 | $0.21_{(0.1)}$ | $0.41_{(0.04)}$ | $1.36_{(0.32)}$ | $1.0_{(0.0)}$ | $1.0_{(0.0)}$ | $4.5_{(0.57)}$ |
| | 100.0 | $0.01_{(0.01)}$ | $0.41_{(0.03)}$ | $0.19_{(0.1)}$ | $0.0_{(0.0)}$ | $0.33_{(0.58)}$ | $0.0_{(0.0)}$ |
| DEMD | 0.001 | $0.38_{(0.05)}$ | $0.44_{(0.04)}$ | $2.85_{(0.39)}$ | $0.37_{(0.01)}$ | $0.25_{(0.0)}$ | $4.78_{(0.32)}$ |
| | 0.01 | $0.38_{(0.05)}$ | $0.45_{(0.03)}$ | $2.85_{(0.38)}$ | $0.38_{(0.01)}$ | $0.26_{(0.0)}$ | $4.82_{(0.32)}$ |
| | 0.1 | $0.38_{(0.05)}$ | $0.44_{(0.03)}$ | $2.83_{(0.39)}$ | $1.0_{(0.0)}$ | $1.0_{(0.0)}$ | $4.33_{(0.42)}$ |
| | 1.0 | $0.35_{(0.05)}$ | $0.43_{(0.03)}$ | $2.54_{(0.39)}$ | $0.93_{(0.12)}$ | $1.0_{(0.0)}$ | $2.67_{(0.17)}$ |
| | 10.0 | $0.26_{(0.05)}$ | $0.39_{(0.02)}$ | $1.35_{(0.25)}$ | $0.68_{(0.28)}$ | $1.0_{(0.0)}$ | $0.9_{(0.47)}$ |
| | 100.0 | $0.0_{(0.0)}$ | $0.41_{(0.03)}$ | $0.26_{(0.02)}$ | $0.0_{(0.0)}$ | $1.0_{(0.0)}$ | $0.0_{(0.0)}$ |

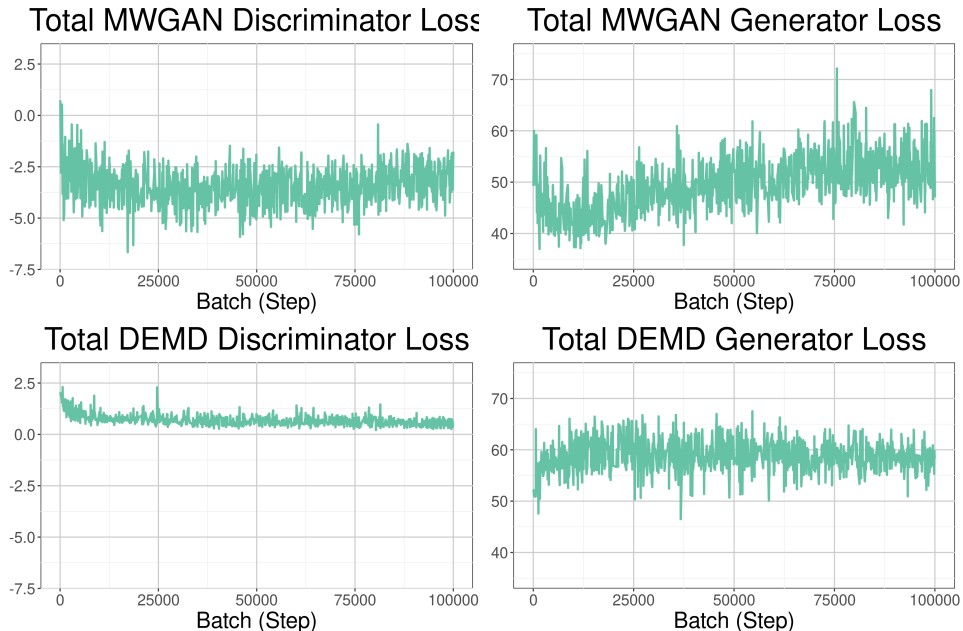

Figure 7: Convergence Plots for MWGAN and DEMD on CelebA Multi-Domain Image Translation.

suggests minimal distributional differences across the groups. For latent feature vector $\ell$ and groups $i/j$, we have

$$\mathcal{MMD} = \| \underset{Z_1 \sim P(\ell))_{\text{group}_i}}{\mathbb{E}} \mathcal{K}(Z_1, \cdot) - \underset{Z_2 \sim P(\Phi(\ell))_{\text{group}_j}}{\mathbb{E}} \mathcal{K}(Z_2, \cdot)\|_{\mathcal{H}} \tag{7}$$

The criterion is defined using a Reproducing Kernel Hilbert Space with norm $\| \cdot \|_{\mathcal{H}}$ and kernel $\mathcal{K}$.

### 7.3.4 MWGAN DETAILS AND ADDITIONAL RESULTS

The MWGAN code `https://github.com/deepmo24/MWGAN` was used under the MIT license extended from the original StarGAN codebase `https://github.com/yunjey/stargan`.

The original paper constructs an inter-domain penalty added to the multi-domain GAN discriminator loss as follows:

$$R_{MWGAN}(f) = \lambda \cdot \left( \sum_i \mathbb{E}_{\tilde{x}^{(i)} \sim \hat{\mathbb{Q}}_i} ||\nabla f(\tilde{x}^{(i)})|| - L_f \right)_+^2 \tag{8}$$

The sampling for expectation in practice is done by interpolating randomly between real data and generated data for each domain. In contrast to this, we directly push the gradient norms computed over samples from each domain to be close using our DEMD regularizer:

$$R_{DEMD}(f) = \lambda \cdot DEMD\left( \left[ ||\nabla f(\tilde{x}^{(i)})|| \right]_i, [i] \right), \tag{9}$$

where $[i]$ indicates the domain associated with the fake samples generated in the same manner for each domain.

**Convergence comparison.** Interestingly we observe that our construction tends to lead to more stable training. In Figure 7, we can see that the variance of the total discriminator and generator loss fluctuate much less. We suspect that this can be directly attributed to the boundedness of the gradients of the DEMD regularizer, specific to the method where we compute the gradients via the dual of the LP: the dual variables are bounded by definition and in this particular LP, are bounded exactly by the number of bins used for discretization. While additional investigation is needed, smoothness has been identified as a desirable property of GAN training (Arora et al., 2017; Chu et al., 2020; Lokhande et al., 2020b; Ravi et al., 2019).

| Model | Blond Hair | Eyeglasses | Mustache | Pale Skin |
|---|---|---|---|---|
| MWGAN Cao et al. (2019) | 49.91 | 45.74 | 45.49 | 38.67 |
| DEMD | 47.29 | 34.43 | 50.69 | 39.60 |

Table 5: Train FID Scores for CelebA Multi-Domain Image Translation.[1]

**Quantitative Results.**    Using the Frechet Inception Distance we quantitatively measure the resulting generative samples. Following the description in Cao et al. (2019), we present the FID scores of our model and theirs in Table 5. Our metric as a drop-in replacement performs comparably.

## 7.4  D-DIMENSIONAL EARTH MOVER'S DISTANCE BACKGROUND AND ALGORITHM

Algorithm 3 describes the greedy algorithm that solves both primal and dual generalized Earth mover's programs, also see (Kline, 2019). The algorithm accepts $d$ distributions (i.e., histograms) $p_1, \ldots, p_d \in \mathbb{R}^n_+$ with $e'p_j = 1$ for all $j \in [d]$. Although the algorithm states that all histograms have the same number of bins, the algorithm can be easily adapted to accept as inputs $p_i \in \mathbb{R}^{n_i}_+$ with $n_i \neq n_j$.

---

**Algorithm 3** EMD Primal/Dual Algorithm

---

**input** $p_j \in \mathbb{R}^n_+$ with $e'p_j = 1$ , $(\forall j \in [d])$
*//iteration index, active indices, primal variables, dual variables*
$k := 0 \in \mathbb{Z}$, $I := 0 \in \mathbb{Z}^d$, $x := 0 \in \mathbb{R}^{n^d}$, $z_j := 0 \in \mathbb{R}^n$, $(\forall j \in [d])$
**while** $I(j) \leq n$, $(\forall j \in [d])$ **do**
$\quad s_k := \min_{j \in [d]} p_j(I(j))$          *// the mass to move*
$\quad x(I) \leftarrow s_k$          *// update the EMD solution*
$\quad p_j(I(j)) \leftarrow p_j(I(j)) - s_k, (\forall j \in d)$          *// shrink the data*
$\quad j^* \leftarrow \arg\min_{j \in [d]} p_j(I(j))$
$\quad I(j^*) \leftarrow I(j^*) + 1$
$\quad k \leftarrow k + 1$
$\quad t_k \leftarrow c(I)$          *// cost of this step*
$\quad$**if** $I(j^*) \leq n$ **then**
$\quad\quad z_{j^*}(I(j^*)) \leftarrow t_k - t_{k-1} + z_{j^*}(I(j^*) - 1)$          *// update the dual solution*
**return** $x$, $(z_1, \ldots, z_d)$, and the objective value $\sum_k s_k t_k$.

---

The algorithm has explicit terminal conditions for the main while loop. In the worst case the number of iterations can be $nd$.

## 7.5  EXTENDED ETHICS DISCUSSION

As discussed in the main paper discussion, the primary application of our proposed construction is to reduce invariance over a particular set of features. In practice, with respect to typical machine learning models and pipelines, this corresponds to minimizing performance difference as measured across subgroups within the data corresponding to a minority or protected subsets of samples or individuals. While the construction can be applied to any existing ML pipeline, we do not claim to provide a catch-all solution for group disparity that may be inherent to the data or exacerbated by the choice of the ML model that is being used. As always, care needs to be taken when working with sensitive data or models which may have disparate impacts on different groups. We point interested readers to the following extensive surveys and references therein regarding various methods and procedures for addressing and dealing with bias and unfairness in ML problems, and the potential danger associated with using models without care: (Mehrabi et al., 2021; Leavy, 2018; O'neil, 2016; d'Alessandro et al., 2017; Rakova et al., 2021).

---

[1]Results presented here computed using the pytorch-fid package with code from the MWGAN repository here: `https://github.com/mseitzer/pytorch-fid`. We were unable to replicate the FID scores provided in the original paper, but expect trends to be relatively similar when compared across different scoring methods.

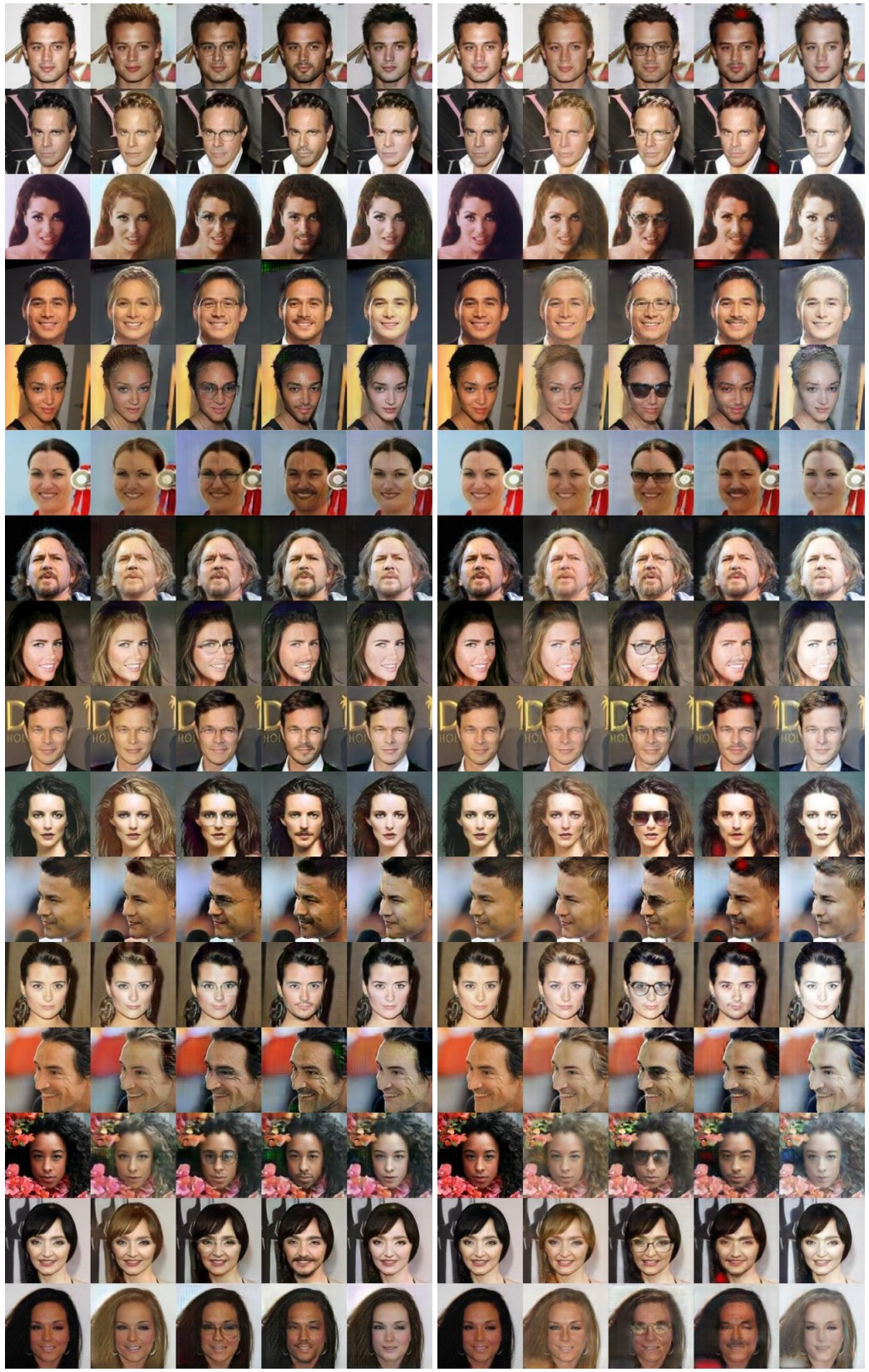

Figure 8: More qualitative results from the multi-domain image translation problem with (Left) Cao et al. (2019), (Right) DEMD (ours). On attributes such as "Blond Hair" and "Eyeglasses", the generated images through our DEMD procedure appear more realistic. On other attributes, the generated images are comparable.

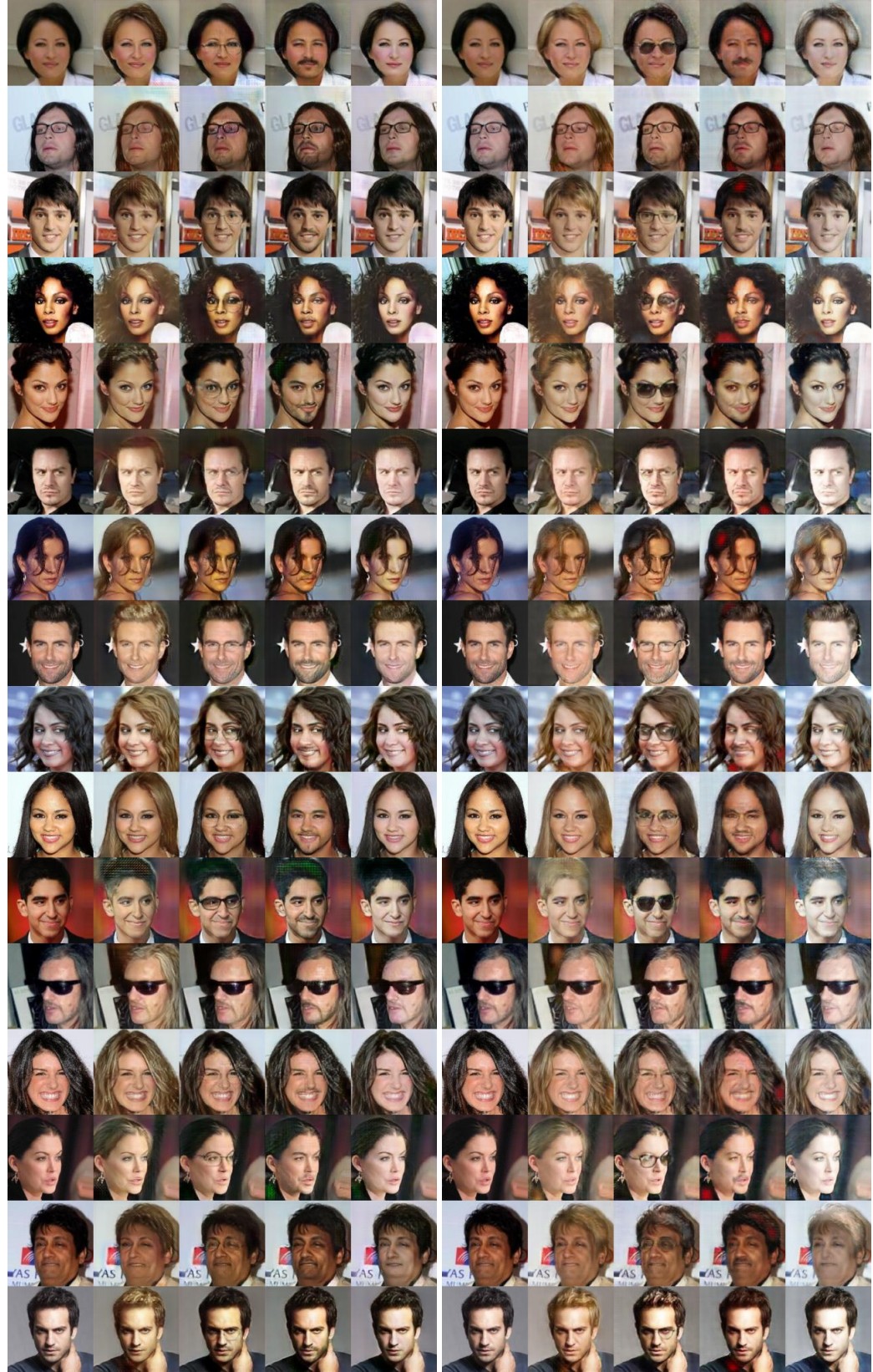

Figure 9: More qualitative results from the multi-domain image translaton problem with (Left) Cao et al. (2019), (Right) DEMD (ours). On attributes such as "Blond Hair" and "Eyeglasses", the generated images through our DEMD procedure appear more realistic. On other attributes the generated images are comparable.

We also note that the final multi-marginal GAN image translation application could be used to generate so-called "deepfakes." While the results of our algorithm are comparable to existing works, we believe that existing methods of identifying deepfakes would work well, and that the methods provided here and in the original paper Cao et al. (2019) would require significant effort to be made practical for much larger scale images.

## 8 ACKNOWLEDGMENTS

This work was supported in part by NIH grants RF1AG059312, RF1AG062336, RF1AG059869, and NSF award CCF 1918211. The work presented here was partially developed while Ronak Mehta, Jeffery Kline, and Glenn Fung were at the Machine Learning Group at American Family Insurance. The authors thank Sathya N. Ravi (UIC) for time reading the paper and providing suggestions.

