# OpenReview forum: "Efficient Discrete Multi Marginal Optimal Transport Regularization"
_ICLR.cc/2023/Conference — ICLR 2023 notable top 25%_

### Official Review · Reviewer_Gjfe · 2022-10-23

**Confidence:** 4
**Correctness:** 4
**Technical Novelty And Significance:** 2
**Empirical Novelty And Significance:** 2
**Recommendation:** 5

**Clarity, Quality, Novelty And Reproducibility:**

This paper has a large room for improvement in clarity and novelty. Please see weakness for more details.

**Strength And Weaknesses:**

The authors should improve the writing of the paper. The overall logic is not smooth. In section 4.1, by using terminologies like “forward”, “backward”, “inner loop”, “outer loop”, the authors already assumed the readers know how the proposed tool is combined with NN, which can actually have many different forms. The definition of this combination is instead introduced in Section 4.2. In Section 4.2, the authors failed to clearly explain why the histograms are needed (although this can be vaguely inferred from some other literature). There are also some minor issues, such as the misusage of \citep and \citet in line 15 of introduction and line 5 of page 3.

I am confused about the contribution of the work. Please correct me if I misunderstand anything: On the theoretical side, the authors prove the second half of Theorem 4.3, i.e., derive the gradient of the objective, which is essentially just the gradient of an LP problem. On the methodology side, the authors apply the derived gradient directly to the backward pass, which is a well-known trick to speed up the computation for problems where there are optimization problems in the forward pass. On the application side, the authors apply the proposed method to fairness, invariant representations, and image translation, where none of the applications are newly set-up. Furthermore, the performance of the proposed method is not as good as Li (2014) and Xie (2017).




**Summary Of The Paper:**

The authors of this work propose a fast algorithm for the computation of the discrete multi-marginal optimal transport problem (MMOT). The algorithm is derived from the dual form of the problem and has an explicit form for the gradient w.r.t. the objective. This makes it much faster than automatic differentiation. The algorithm is applied to fairness problems and image translation, demonstrating its effectiveness in both domains.

**Summary Of The Review:**

Although the proposed method can be useful for some applications, I do not think this work has enough novelty to be accepted.

I would encourage the authors to theoretically or empirically explore the complexity of the so-called “outer loop” under some simplified setting. This would make the work more solid and complete.

---

> ### Author Response · Authors · 2022-11-09
> **Gjfe Review - Author Response Part 1**
>
> Thank you for the detailed comments! Your comments have absolutely helped in making parts of our paper clearer, and as mentioned in specific responses below, we're more than happy to make these changes to increase clarity.
>
> > The authors should improve the writing of the paper. The overall logic is not smooth. In section 4.1, by using terminologies like “forward”, “backward”, “inner loop”, “outer loop”, the authors already assumed the readers know how the proposed tool is combined with NN, which can actually have many different forms. The definition of this combination is instead introduced in Section 4.2.
>
> We appreciate your detailed feedback regarding the structure and clarity of these specific sections. We agree that in some sections, we assumed a certain level of reader familiarity; our original target audience was one that may already be familiar with optimal transport methods within machine learning. We agree that some mild restructuring in Sections 4.1 and 4.2 will increase the clarity of our work, specifically moving some of the discussion from the top of Section 4.2 to the beginning of Section 4.1. This can be reflected in an updated version of the paper. We welcome any other specific suggestions you might have.
>
> >  In Section 4.2, the authors failed to clearly explain why the histograms are needed (although this can be vaguely inferred from some other literature).
>
> Our discussion under the paragraph heading “Need for Histogramming” was set up to address this question. Looking back, if there is anything here that is unclear or needs to be expanded, we would be happy to do so. There is some mild additional discussion in the Appendix also, but we would much appreciate clarification on what you feel is poorly explained or argued.
>
> > There are also some minor issues, such as the misusage of \citep and \citet in line 15 of introduction and line 5 of page 3.
>
> Thank you for pointing this out, we will fix this typo! We would greatly appreciate it if you could point out any other minor issues which we are happy to adjust also.

---

> ### Author Response · Authors · 2022-11-09
> **Gjfe Review - Author Response Part 2**
>
>
> > I am confused about the contribution of the work. Please correct me if I misunderstand anything: On the theoretical side, the authors prove the second half of Theorem 4.3, i.e., derive the gradient of the objective, which is essentially just the gradient of an LP problem.
>
> The first part of Theorem 4.3, discussing the gradient, is indeed covered by prior work, which we provide references for in the first few lines of the proof in the Appendix.
>
> The second part of Theorem 4.3 concerns affine shifts of dual solutions, and specifically builds upon the “zero-sum” result of Theorem 3.1 in (Kline, 2019). We agree that we can expand this point by adding a comment in the proof. We briefly touch on it in Remark 4.4; in practice dual solutions found by different methods do not agree. This result provides an effective way to reconcile these different solutions. This property was particularly important during our experimental development, where different methods (NumPy, Torch, etc.) result in differing gradients. We discuss some of this in the proof within the Appendix, and would be happy to include more of it within the main paper to better demonstrate the value of our theoretical contribution.
>
> > On the methodology side, the authors apply the derived gradient directly to the backward pass, which is a well-known trick to speed up the computation for problems where there are optimization problems in the forward pass.
>
> We hope that the reviewer will appreciate the contrast between this particular methodological approach compared to unrolling methods typical within the OT literature. Among those approaches, many of the problem formulations do not lend themselves to simple gradient derivations. Within this space, we believe that our observation is different and novel and, while simple, is valuable as a viable (and previously unexplored) alternative to entropic regularization approaches.
>
> > On the application side, the authors apply the proposed method to fairness, invariant representations, and image translation, where none of the applications are newly set-up. Furthermore, the performance of the proposed method is not as good as Li (2014) and Xie (2017).
>
> The goal of these application experiments was not to propose novel experiments, but rather to leverage multi-marginal optimal transport to enforce distribution constraints in a variety of relevant application settings. Among the several applications, DEMD outperforms the baseline in obtaining fairer representations (Table 1) and a higher quality image translation (Table 5). The baselines adopted in the harmonization experiments are not straightforward -- recall that  Li (2014) and Xie (2017) directly optimize for the ADV and MMD measures respectively. Our results in Table 2 suggest that DEMD is an effective harmonization technique despite not directly optimizing the invariance measures at all.
>
> > I would encourage the authors to theoretically or empirically explore the complexity of the so-called “outer loop” under some simplified setting. This would make the work more solid and complete.
>
> We believe there is a small confusion here, so we want to use this opportunity to clarify.
>
> The loop terminology was used only to simplify understanding but yes, if taken literally, may raise a doubt regarding the influence on the complexity of the outer loop. The reviewer will see that we are solving the inner problem exactly, where the inner “loop” is really a fixed function of the input (via the forward pass). It is more convenient to view it as a first order oracle, so a complexity analysis of the outer loop would remain agnostic. The outer loop is the only iterative optimization, and known results would apply (Allen-Zhu et al, 2019 [1]) depending on the Lipschitz constant of the network.
> Since the inner module involves a fixed iteration forward pass O(nd) and the backward pass is O(1), the empirical complexity of the outer loop will depend on the DNN architecture chosen. Please let us know if we interpreted the question correctly. Thanks.
>
> [1] Allen-Zhu, Zeyuan, Yuanzhi Li, and Zhao Song. "A convergence theory for deep learning via over-parameterization." International Conference on Machine Learning. PMLR, 2019.

---

> > ### Comment · Reviewer_Gjfe · 2022-11-18
> > **Gradient derivations are common in the OT literature**
> >
> > To my understanding, explicit gradient derivation is not uncommon, even in OT literature. See [1] [2] [3].
> >
> > [1] Luise G, Rudi A, Pontil M, et al. Differential properties of sinkhorn approximation for learning with wasserstein distance[J]. Advances in Neural Information Processing Systems, 2018, 31.
> >
> > [2] Xie Y, Dai H, Chen M, et al. Differentiable top-k with optimal transport[J]. Advances in Neural Information Processing Systems, 2020, 33: 20520-20531.
> >
> > [3] Cuturi M, Teboul O, Niles-Weed J, et al. Supervised quantile normalization for low rank matrix factorization[C]//International Conference on Machine Learning. PMLR, 2020: 2269-2279.

---

> > > ### Author Response · Authors · 2022-11-19
> > > **Contrasting Gradient Derivations**
> > >
> > > We absolutely agree that explicit gradient derivation is common. However, the work highlighted derives explicit gradients _after approximation_. In each case (pg. 3 eq. 6 in [1], Section 2.3, eq. 5 in [2], and Section 2.1, Eq. (P-RegOT) in [3]), an entropic regularization is _required_, and Sinkhorn iterations are used in an “unrolled” manner. In contrast, our gradient computation is of the _exact_ original problem formulation, and our forward pass and gradients terminate in _exactly_ $nd$ and $O(1)$ iterations, respectively.
> > >
> > > We have updated and combined Section 4.1 and 4.2 for better clarity, and have included those references with text clarifying this distinction between existing work and our paper. We appreciate the suggestions. Thank you!

---

> ### Author Response · Authors · 2022-11-14
> **Have we addressed your concerns?**
>
> Dear Reviewer Gjfe,
>
> Please let us know if our responses have addressed your concerns, and if there are any other additional questions you may have which we might be able to answer during this discussion phase. When you are satisfied with our responses we will upload a revised version of the paper.
>
> Thank you so much for your time and your feedback!

---

### Official Review · Reviewer_RRc2 · 2022-10-25

**Confidence:** 4
**Clarity, Quality, Novelty And Reproducibility:** The paper is clearly written, of high…
**Correctness:** 3
**Technical Novelty And Significance:** 3
**Empirical Novelty And Significance:** 3
**Recommendation:** 8

**Strength And Weaknesses:**

Strengths:
- OT is a method of increasing importance in the machine learning literature.
- Multi marginal OT is potentially important but hampered by significant computational costs that scale poorly with additional dimensions.
- The theory clearly demonstrates the computational opportunity, which is nicely confirmed experimentally.
- The empirical demonstrations show that the approach can be deployed on interesting problems.

Weaknesses:
- I did not find any particularly notable weaknesses of the paper. While the setting is clearly a restriction of the broader MMOT problem, the authors convincingly argue that it is not as much a restriction as one might think, and the computational gains are significant and worthy of the compromise.

**Summary Of The Paper:**

The paper introduces a novel, efficient method for computing multimarginal optimal transport plans. The approach applies to cases with Monge cost, which while a special case, the authors argue is relatively general. A generalized earth-mover distance is introduced, which is linear in the dimensions and bins in theory, and show that in practice can lead to orders of magnitude speed up relative to existing methods. Empirical results are demonstrated in fairness cases as well as image translation.

**Summary Of The Review:**

The paper introduces generalized earth mover distance which enables efficient computation of MMOT plans. The approach represents a significant advance over existing methods for computing MMOT plans, and efficiency gains are clear theoretically and practically. The approach is demonstrated on domains of interest. This is a nice contribution.

---

> ### Author Response · Authors · 2022-11-08
> **Thank You!**
>
> Dear reviewer, thank you for your kind words! We appreciate that you found our work clear, novel, and of high quality, and believe it to be a nice contribution.
>
> > Correctness 3: Some of the paper’s claims have minor issues. A few statements are not well-supported, or require small changes to be made correct.
>
> If you could kindly point us to specific statements that you feel can be clarified more, we would be happy to identify ways in which the argument could be improved!

---

> ### Author Response · Authors · 2022-11-14
> **Suggestions for improvements**
>
> Reviewer RRc2,
>
> Please let us know if we can address your concern regarding correctness, and if there are any other additional questions you may have which we might be able to answer during this discussion phase.
>
> After discussion, we will upload a revised version of the paper.
>
> Thank you so much for your time and your feedback!

---

### Official Review · Reviewer_7jEQ · 2022-11-04

**Confidence:** 3
**Correctness:** 4
**Technical Novelty And Significance:** 2
**Empirical Novelty And Significance:** 2
**Recommendation:** 6

**Clarity, Quality, Novelty And Reproducibility:**

The paper is clear and well structured.
To the best of my knowledge, all claims have been referenced to existing work or supported with theoretical or empirical results in the paper.
Narrowing down the ground cost to be Monge, which yields fast iterations, in MMOT problems seems novel to me.
The paper also details the experiment setup, results seem reproducible.

**Strength And Weaknesses:**

Strength:
Efficient algorithms for solving MMOT is an open field, especially a general one. This paper finds a sweep spot in the ground metric where the problem can be solved in linear time and differentiable to transportation map.

Weaknesses:
I haven't found major weakness of the paper. The theoretical contribution to MMOT seems marginal though.

**Summary Of The Paper:**

The paper introduces a new algorithm to compute MMOT. It narrows down the ground cost to deal with a specific form of MMOT. That results in fast computation while still maintains its generalization to normal use cases. Simplification is supported with theoretical proof. Authors also insert a few tricks to speed up and differentiate the iterations. Empirical results show improvement over different tasks after applying the proposed algorithm in network training.

**Summary Of The Review:**

I'm leaning toward acceptance.
The paper is well written, theoretically solid, and it offers new insights in accelerating the computation of MMOT.
Experiments also show several use cases of the new algorithm.

---

> ### Author Response · Authors · 2022-11-08
> **Thank you! and a Note on Significance**
>
> Dear reviewer, Thanks for appreciating our work, and the positive feedback on the organization and presentation! As you noted, we target the MMOT problem under reasonable conditions that lead to fast and feasible solutions.
>
> > Weaknesses: I haven't found major weakness of the paper. The theoretical contribution to MMOT seems marginal though.
>
> Our goal was to identify a slightly narrower MMOT setting that allows much faster practical solutions. The MMOT problem has not been studied under Monge costs and our main hypothesis was that this choice will not be restrictive and allow significant benefits.
>
> Our key findings are that
> 1) the Monge cost enables a much faster solver for MMOT and
> 2) that the gradient is readily available for our MMOT model
> These are interesting insights for MMOT which we believe offer benefits and a sensible option to consider in problem settings where barycenters are useful.
>
> Further, the characterization of affine shifts among dual solutions is also important, and was not obvious at first. Without this result, we would not be able to reconcile differing solutions from the same problem instantiation. Some of this discussion is in the Appendix, but we would be more than happy to include more of it within the main paper if suggested.
>
> We appreciate the large amount of existing and ongoing theoretical work behind classical MMOT problems, and acknowledge in our discussion section that there are various other problems that could be addressed. We hope that with this paper we can offer a self-contained, complete starting point from which future work can effectively build on. We expect that our practical algorithms, alongside our easy to use code, will reduce the barrier to future development.

---

> ### Author Response · Authors · 2022-11-14
> **Have we addressed your concerns?**
>
> Reviewer 7jEQ,
>
> Please let us know if our response has addressed your concerns, and if there are any other additional questions you may have which we might be able to answer during this discussion phase.
>
> When you are satisfied with our responses we will upload a revised version of the paper.
>
> Thank you so much for your time and your feedback!

---

### Author Response · Authors · 2022-11-19
**Addressing Concerns and Rebuttal Revisions**

As mentioned in our response to Reviewer Gjfe, we have updated and combined Section 4.1 and 4.2 for better clarity, including better reference to and distinction from related work. These changes for clarity have been highlighted in blue.

We appreciate all reviewer comments and suggestions, hope we have addressed your other concerns, and are happy to clarify and discuss any other points by which we may improve the quality of our submission. Thank you!

---

### Decision · Program_Chairs · 2023-01-20

**Decision:**

Accept: notable-top-25%

**Justification For Why Not Higher Score:**

see meta-review.

**Justification For Why Not Lower Score:**

see meta-review.

**Metareview: Summary, Strengths And Weaknesses:**

This work proposes a novel method for solving discrete multi-marginal optimal transport problems, and presents its advantages for automatic computation of gradients that can be used in model training.

The reviewers and I agree that this is a very good submission that presents new, interesting ideas, in a well-exposed manner.

**Note From Pc:**

if the above contains the word "oral" or "spotlight" please see: "oral" presentation means -> notable-top-5% and "spotlight" means -> notable-top-25%. As stated in our emails, we are disassociating presentation type from AC recommendations